# Humoral Response after Vaccination with Half-Dose of BNT162b2 in Subjects under 55 Years of Age

**DOI:** 10.3390/vaccines9111277

**Published:** 2021-11-04

**Authors:** Krzysztof Lukaszuk, Amira Podolak, Grzegorz Jakiel, Jolanta Kiewisz, Izabela Woclawek-Potocka, Aron Lukaszuk, Lukasz Rabalski

**Affiliations:** 1Invicta Research and Development Center, 81-740 Sopot, Poland; luka@gumed.edu.pl (K.L.); lukaszuk.aron@gmail.com (A.L.); 2Department of Obstetrics and Gynecology Nursing, Medical University of Gdansk, 80-210 Gdansk, Poland; 3The Center of Postgraduate Medical Education, 1st Department of Obstetrics and Gynecology, University of Gdansk, 01-004 Warsaw, Poland; grzegorz.jakiel1@o2.pl; 4Department of Human Histology and Embryology, Medical Faculty, Collegium Medicum, University of Warmia and Mazury in Olsztyn, 10-082 Olsztyn, Poland; jolanta.kiewisz@uwm.edu.pl; 5Department of Gamete and Embryo Biology, Institute of Animal Reproduction and Food Research, Polish Academy of Sciences, 10-748 Olsztyn, Poland; i.woclawek-potocka@pan.olsztyn.pl; 6Laboratory of Recombinant Vaccines, Intercollegiate Faculty of Biotechnology, University of Gdansk, 80-307 Gdansk, Poland

**Keywords:** SARS-CoV-2, COVID-19, mRNA vaccine

## Abstract

In the context of the ongoing COVID-19 pandemic, using a half-dose schedule vaccination can help to return to normalcy in a cost-efficient manner, which is especially important for low and middle-income countries. We undertook a study to confirm that in adults up to 55 years old, the humoral response to the half-dose (15 µg, 35 participants between 18 and 55 years old) and to the recommended dose (30 µg, 155 participants) in the two-dose three-week interval schedule would be comparable. Antibody levels were measured by the Elecsys Anti-SARS-CoV-2 S assay (Roche Diagnostics, upper detection limit: 2570 BAU/mL) on the day of dose 2 of the vaccine and then 8–10 days later to assess peak response to dose 2. The difference in proportions between the participants who did and did not exceed the upper detection limit 8–10 days after dose 2 was not statistically significant (*p* = 0.152). We suggest that a half-dose schedule can help to achieve widespread vaccination coverage more quickly and cheaply.

## 1. Introduction

In the context of the ongoing COVID-19 pandemic, rapid vaccination appears to be the only path to the return to normalcy. Several manufacturers began work on their vaccines in January 2020. In November 2020, Pfizer/BioNTech and Moderna announced the first results from the Phase 3 clinical trials showing high vaccine efficacy for the doses that were selected for testing [1,2]. Understandably, the speed of development was of prime importance; the company’s and public’s funding were at stake. Therefore, successful approval of the selected vaccine variants and doses was of prime importance. The ultimately produced novel mRNA vaccines were characterized by low production costs, high efficiency in triggering the expected humoral response, and tolerable side effects after application [1,2,3,4]. 

However, both undersupply and the high selling prices in low- and middle-income countries require broadening our perspective [5]. The World Health Organization (WHO) highlights the importance of urgent action to boost supply and ensure equitable access for every country [6]. Even though the U.S. and the G7 nations are committing support for global vaccination programs through COVID-19 Vaccines Global Access (COVAX), only 2.5% of the population in low-income countries have received at least one dose of the SARS-CoV-2 vaccine [7], and less than 1% are fully vaccinated [6,8]. This program can vaccinate approximately 20% of the population in participating countries by the end of 2021 [9]. This falls far short of the goal of achieving global herd immunity. 

Recently, WHO, with the support of the Strategic Advisory Group of Experts (SAGE) in Immunization and its COVID-19 Working Group, published a statement encouraging research in the area of vaccine reduction [10]. Such a dose-reduction strategy has led to the successful vaccination of millions of people in Africa and South America against yellow fever [11]. Thus, as there is an urgent need to re-evaluate the dosing schedules of the available vaccines to achieve the goals of global SARS-CoV-2 elimination or at least the state of cohabitation, we decided to conduct a pilot study giving people under 55 half the recommended standard dose. We aimed to show that in adults up to 55 years old, the humoral response to the half-dose would be comparable with the response to the recommended dose when vaccinated in the prime-boost three-week interval schedule. Moreover, we took into consideration prior COVID-19 infection.

## 2. Materials and Methods

### 2.1. Ethical Policy

Ethical approval was received from the Ethics Committee at the Gdansk Regional Medical Board (No KB-14/21). All participants gave written informed consent for providing blood samples.

### 2.2. Vaccination

The study group received half the recommended dose (i.e., 15 µg, HD) of the Pfizer/BioNTech BNT162b2 (BNT162b2) vaccine twice, at an interval of 21 days. The control group was vaccinated using the recommended dose of 30 µg (RD) of BNT162b2, separated by a 21-day interval.

### 2.3. Participants

The HD group consisted of 42 people. Among the HD group, 7 subjects reported prior COVID-19 infections confirmed with a PCR test, and 2 had a positive result with a measurable antibodies level before the first dose of the vaccine. Those participants (*n* = 9) were included in the subgroup of participants with prior COVID-19 infection (HD-COVID-19-positive). The remaining 33 subjects were classified as HD-COVID-19-negative. In 5 cases with COVID-19 confirmed by PCR assay, we retrieved archived material for virus sequencing. 

The RD group consisted of 188 healthcare professionals who were vaccinated as part of the national vaccination program. Among them, 31 individuals had a history of COVID-19 infection before vaccination (RD-COVID-19-positive). The participants who reported no history of COVID-19 but had a positive antibody test before vaccination (*n* = 5) were also included in the RD-COVID-19-positive subgroup. The RD-COVID-19-negative subgroup included 152 subjects. In 27 cases from the RD-COVID-19-positive subgroup, we were able to obtain sample material for virus sequencing.

The characteristic of attendees and the inclusion and exclusion criteria are presented in Table 1 and Table 2, respectively.

The patients’ recruitment and study protocol are presented in Figure 1.

### 2.4. SARS-CoV-2 Antibodies Measurement

All participants had the initial measurement of their level of SARS-CoV-2 antibodies on the first day of the vaccine dose.

The subjects included in the COVID-19-positive subgroups were retested 8 days after the first dose. The 8-day interval was chosen, based on our previous studies, where we found that the peak patient response to the second dose is achieved on day 8 after vaccination (unpublished data). Other studies [12] show that in individuals who had a prior SARS-CoV-2 infection, the response to the first vaccine dose is similar to the reaction to the second vaccine dose in those without an infection history. The participants who were not affected with COVID-19 infection were examined 21 days after the first dose (i.e., on the day of the second vaccine dose), and then 8 days after the second dose, when the expected humoral response is highest.

### 2.5. Assay Characteristics

Antibody levels were measured by the Elecsys Anti-SARS-CoV-2 S assay (Roche Diagnostics, Switzerland) (Roche). This assay is intended for the quantitative determination of IgG and IgM high-affinity antibodies against SARS-CoV-2 spike (S) protein receptor-binding domain (RBD) in human serum and plasma. 

This is an automated assay certified for in vitro diagnostic (IVD) clinical use. The assay is based on the electro-chemiluminescence immunoassay (ECLIA) method and the double-antigen sandwich assay test principle. Two-point calibration is recommended for each new reagent lot and whenever the control results exceed the specified limits. It was initially standardized against the manufacturer’s internal standard, and then from 12 January 2021, standardized against WHO International Standard NIBSC20-136. The assay requires 12 μL of serum collected using standard testing tubes. Testing time is 18 min.

The test results are reported in units per mL (U/mL) and can be converted to the WHO standard, defined as Binding Antibody Units per mL (BAU/mL), by multiplying the Roche result by 1.029, according to the manufacturer. As suggested previously [13], the identifiers should be added to the unit designation to show what is being measured. Therefore, for the Roche assay, we described the unit as BAU_RBD-IgG+IgM_/mL. The assay’s limit of detection (LoD) is 0.36 BAU_RBD-IgG+IgM_/mL, the limit of quantification (LoQ) is 0.41 BAU_RBD-IgG+IgM_/mL. The measurement range is 0.41–257 BAU_RBD-IgG+IgM_/mL, and up to 2570 BAU_RBD-IgG+IgM_/mL for 10-fold diluted samples.

### 2.6. Sequencing

Viral RNA was extracted from oropharyngeal and nasal swabs collected and stored in viral transport medium in consonance with WHO and CDC recommendations [14,15], using a CoV RNA kit (A&A Biotechnology), according to the manufacturer’s instructions. Nucleic acid amplification testing (NAAT) was performed using a reverse transcriptase real time-PCR kit (genesig^®^ by Primerdesign Ltd., Southampton, UK), CE-IVD marked, for detecting the RdRp gene. RT-PCR is considered the gold standard for the identification of the SARS-CoV-2 virus [16,17,18]. According to different studies, the Coronavirus COVID-19 genesig^®^ Real-Time PCR assay exhibits 95–100% clinical sensitivity and 100% clinical specificity [19,20]. The Foundation for Innovative New Diagnostics (FIND), in their comparative LOD study, reports that the test achieved the highest LOD banding (1–10 copies/reaction) [19]. Moreover, the test shows no cross-reactivity with other confirmed respiratory viral infections, such as influenza virus type A, RSV A and B, and rhinovirus [21]. To validate the obtained results, positive and negative controls were included every time the test was conducted. Furthermore, internal RNA extraction control was used as a positive control for the extraction process. Successful qPCR of the exogenous source of the RNA template, spiked into samples during RNA extraction, additionally indicated that the PCR inhibitors were not present at a high concentration. 

ARTICv3 amplicon generation, followed by an Oxford Nanopore Technology MinION run, was performed for COVID-19 positive patient samples. RNA was converted to cDNA with the RT-PCR technique, using the LunaScript^®^ RT SuperMix Kit (New England BioLabs Inc., Ipswich, UK). Then, the obtained cDNA was amplified to prepare libraries for nanopore sequencing. Amplification was performed using Q5^®^ Hot Start High-Fidelity 2X Master Mix (New England BioLabs Inc., Ipswich, UK) for two reactions per sample with thermocycling conditions customized to primers’ melting temperatures. The concentration of the PCR products was measured with the Qubit 1x dsDNA HS (High-Sensitivity) Assay Kit (ThermoFisher, Waltham, MA, USA). All samples were barcoded and multiplexed together. Then, the AMII adapter ligation reaction was performed using Quick T4 DNA Ligase (New England BioLabs Inc., Ipswich, UK). Final library dilution was prepared with sequencing buffer and loading beads (Oxford Nanopore Technologies, Oxford, UK). To monitor contamination, NRT controls were processed through reactions. Reads were base-called, de-barcoded, and trimmed to remove the adapter, barcode, and PCR primer sequences. ARCTIC pipeline software was used to generate the SARS-CoV-2 genome (ARTIC-nCoV-bioinformaticsSOP-v1.1.0).

### 2.7. Statistical Analysis

Statistical analyses were performed using R packages (tidyverse). The Wilcoxon signed-rank test was used to assess the differences in mean antibody levels.

## 3. Results

### 3.1. The Comparison of Mean Antibody Level in the Serum of COVID-19 Positive Subjects

There were no statistical differences between mean antibody levels in the serum of participants of the HD-COVID-19-positive (50.8 BAU_RBD-IgG+IgM_/mL; *n* = 9) and RD-COVID-19-positive group (44.1 BAU_RBD-IgG+IgM_/mL; *n* = 36) before vaccination. At this point in time, more than 80% of participants had antibody levels in the range of 10–100 BAU_RBD-IgG+IgM_/mL (Figure 2A).

Most test results obtained on day 8 and day 21 after the first dose of the vaccine exceeded the upper detection limit of the Roche assay. The breakdown of the results from day 8 after the first vaccine dose is shown in Figure 2B.

### 3.2. The Mean Antibody Level in the Serum of COVID-19 Negative Subjects

Participants of the HD- and RD-COVID-19-negative subgroups had a negative antibody test result before the first dose of the vaccine. Their first post-vaccination measurement time-point was 21 days after the first dose of the vaccine (i.e., on the day of the second dose). The mean antibody level in the HD-COVID-19 negative group was 82.2 BAU_RBD-IgG+IgM_mL, and in the RD-COVID-19 negative group, was 113 BAU_RBD-IgG+IgM_/mL (Figure 3). The detection limit of the Roche kit was exceeded 8–10 days after the second dose in 58% of the participants in the HD-COVID-19 negative group, and 70% of the RD-COVID-19 negative group. The difference in proportions between the participants who did and did not exceed the detection limit 8–10 days after dose 2 was not statistically significant (*p* = 0.152). Mean antibody levels for participants who did not exceed the 2570 _RBD-IgG+IgM_mL detection limit were 1300 _RBD-IgG+IgM_/mL (*n* = 14) and 1700 BAU_RBD-IgG+IgM_/mL (*n* = 45) for the HD-COVID-19 negative and RD-COVID-19 negative subgroups, respectively 

## 4. Discussion

The current study compares the humoral response to the half-dose and recommended dose of the vaccine administrated to adults up to 55 years old. A strong humoral response was observed 8 days after the first dose of vaccine in the group of subjects with previous COVID-19 infection. Participants in both the RD and HD group who did not have COVID-19 disease, showed a weaker reaction in response to the first dose of the BNT162b2 vaccine, and had a peak humoral response 8 days after the second vaccine dose with the majority exceeding the upper test detection limit. We observed no differences between the mean antibody levels in the serum of the RD and HD groups of COVID-19 negative subjects. It seems that there were no differences in humoral response between the groups vaccinated with the half-dose and the recommended dose.

In addition, pre-vaccination results showed that 3% of subjects did not know they were infected with SARS-CoV-2.

Pfizer/BioNTech, during phase 1 of the vaccine preparation trial, excluded all participants with prior infection, confirmed not only by positive PCR but also serologically [1,3]. While in the Moderna vaccine trial, subjects with a history of disease were allowed to participate [2]. It should be kept in mind that including such participants in the analysis may bias the results. On the other hand, their participation in such a study provided proof of a much higher immunological response in previously infected individuals due to the previous contact with the pathogen [12,22,23].

Interestingly, the humoral response after the first dose in the HD group was comparable to the response observed in the RD group. Both immune reactions did not achieve a level that significantly reduces the risk of infection. However, the priming dose of the vaccine elicits a response that sets the stage for a robust and faster reaction to subsequent contact with the antigen. Therefore, individuals infected with a low dose of the virus are capable of a response that prevents the virus from breaking through the immune system, and thus have a significantly lower risk of severe infection or death [24]. Instead, they provide a transmission link for the disease. However, exposure to a large dose of the virus after a single dose of the vaccine may not give the infected person time for a secondary response. Hence, frequent cases of severe infection after only one dose of the vaccine are observed.

We have showed that administration of a lower dose of vaccines in a group of people younger than 55 years old appears to provide sufficient protection for most people at doses of the virus to which people are exposed, on average. Various vaccine dosing schedules were evaluated during phase 1 of vaccine trials. 

Phase 1 of the Pfizer trial compared the immune response with various vaccine doses between participants 18–55 and 65–85 years of age. Each group included 15 participants, with three receiving the placebo. Their response was compared with the results obtained from the serum of COVID-19 convalescent donors (serum was obtained at least 14 days after diagnosis was confirmed by a PCR test). For each tested vaccine dose, the immune response measured by the antibody and neutralization assays was lower in the older group. It appears that the 30 µg dose was selected as it was the only dose in the 65–85 group where the response of the trial participants exceeded the COVID-19 convalescent results. However, the results for the younger group after the second dose of the vaccine exceeded the COVID-19 convalescent results in all tested vaccine doses at 10 µg, 20 µg, and 30 µg [1,3,4,25,26].

In our study, all participants with a history of prior SARS-CoV-2 infection had antibody assay results above the detection range of the test after the first dose of the vaccine, in both the half-dose and the recommended dose groups. Other studies have also shown that vaccine recipients with prior infection react to the first vaccine dose similarly to how those who were not infected respond to the second dose [12].

Moderna’s phase 1 trial of the mRNA-1273 vaccine in the 18–55-year-old population included 45 participants divided into three groups for the tested doses of 25 µg, 100 µg, and 250 µg [27]. The additional expanded trial of adults older than 55 years of age, had 40 participants divided into four groups based on age (56–70 and ≥71 years old) and dose (25 µg and 100 µg). In both parts of the trial, participants were not screened for a history of SARS-CoV-2 infection. It is clearly noted in the expanded portion of the trial that includes participants over 55 years of age, that antibodies against SARS-CoV-2 RBD were found in the serum before the vaccination [2]. Therefore, the mean response in each group depended not only on the vaccine dose but also on the number of participants with prior infection. 

Moreover, 7 months after the first administration of the 25 µg dose in Phase 1 of the Moderna trial, Mateus et al. [28] examined vaccine-specific CD4+ T cells, CD8+ T cells, and binding antibody and neutralizing antibody responses. Six months after the second dose, nearly all of the 35 participants had neutralizing antibodies blocking the virus from infecting cells, and levels of both antibodies and T cells were comparable to those induced by natural infection. In addition, the Moderna mRNA-1273 vaccine is currently undergoing studies using a half-dose for booster doses, but data are not yet available [10].

The AstraZeneca phase 1 trial focused on comparing a single dose schedule with the prime-boost schedule (COOV1). Exclusion criteria included prior confirmed COVID-19 infection, but as the serology testing was not available, some participants included in the study had likely undergone the infection asymptomatically. It was noted in the study results that some participants exhibited high levels of neutralizing antibodies at baseline, thus skewing the results obtained. In the case of AstraZeneca, the lower vaccine dose was tested in phase 2 and phase 3 trials, as a result of the difference in the measurement of viral particles used in the manufacturing process. Due to the discovered differences, additional study groups were formed for participants who received the lower dose (LD) as the first dose, and the standard dose (SD) as the second dose. The interim trial analysis has shown the efficacy to be higher in the LD/SD group than in the SD/SD group [29,30]. 

The limitation of our study is the lack of evaluation of antibody levels in a longer follow-up period. However, when compared to the Moderna trial, it is promising that the protection may be the same as the recommended dose. Still, even if the immune responses to the low-dose strategy may only be moderately effective, it could be worth speeding up the pace of vaccination as fractionated doses could provide a feasible solution that extends limited supplies of vaccines against COVID-19, which is a major challenge for low- and middle-income countries [31]. According to the Wiecek et al. [32] modelling study, the low-dose strategy would reduce infections and COVID-19 linked deaths more than current policies. For the mRNA-1273 vaccine, the model proposed by Khoury et al. [33] suggests that if vaccine efficacy at the complete dose is 95%, a reduction in the dose that led to as much as a halving in the post-vaccination geometric mean titer could still be in the range of 85–90%. To our knowledge, there is one trial ongoing in Belgium comparing the lower dose version (20 µg) of the Pfizer/BioNTech vaccine versus the standard dose in healthy adults up to 55 years of age. The study completion date is estimated to be 30 September 2022 [34].

Another limitation is that we do not know how high antibody levels would need to be to sufficiently have the effect of neutralizing the virus and blocking it from entering the host cell and multiplying. However, the data already available show that it correlates with measured levels of binding antibodies [35,36]. Although research on the WHO International Standard introduced the goal of standardizing the results obtained with different SARS-CoV-2 antibody assays and correlating their level with the neutralizing effect, this has not been confirmed in previous studies [13,23]. The dangers of new mutations escaping the neutralizing properties of antibodies formed both after COVID-19 and after vaccination, should be the impetus to vaccinate the maximum possible proportion of the population as soon as possible, to obtain the maximum possible neutralizing antibody levels in that population. This is to not only reduce the disease mortality but also to limit its spread and minimize the time and population space for the virus to mutate [37,38].

## 5. Conclusions

We believe that using a half-dosage schedule could be one way to increase vaccine supply and to accelerate population-level vaccination coverage in low- and middle-income countries to reduce mortality, as was demonstrated in the past with the yellow fever vaccine.

The study needs to be conducted on a larger group with a longer follow-up period. On the other hand, the results of the study that administered a third dose (i.e., booster), may nullify the need for additional studies and point us in the direction of vaccinating those receiving half a dose with a booster after a specified interval. The third dose may, presumably, be even less than half of the currently administered dose.

## Figures and Tables

**Figure 1 vaccines-09-01277-f001:**
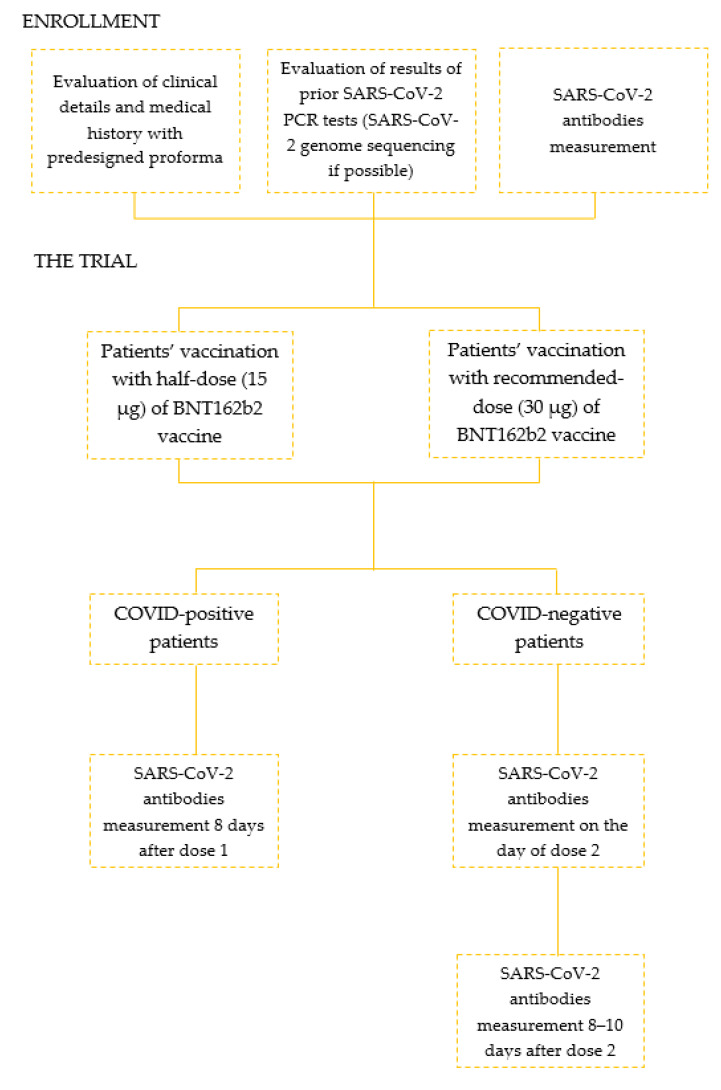
The flow-chart of patients’ recruitment and study protocol.

**Figure 2 vaccines-09-01277-f002:**
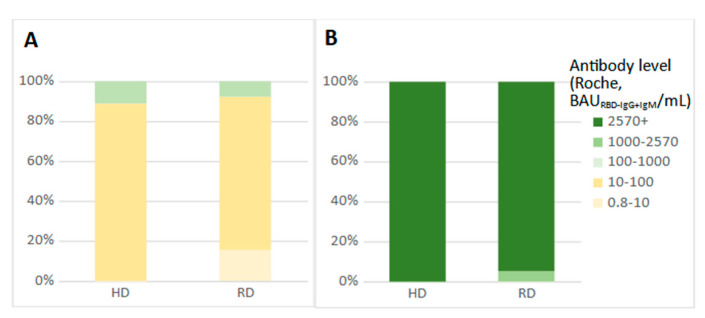
Comparison of antibody levels in the HD- and RD-COVID-19-positive subgroups before vaccination (**A**) and 8 days after the first vaccine dose (**B**).

**Figure 3 vaccines-09-01277-f003:**
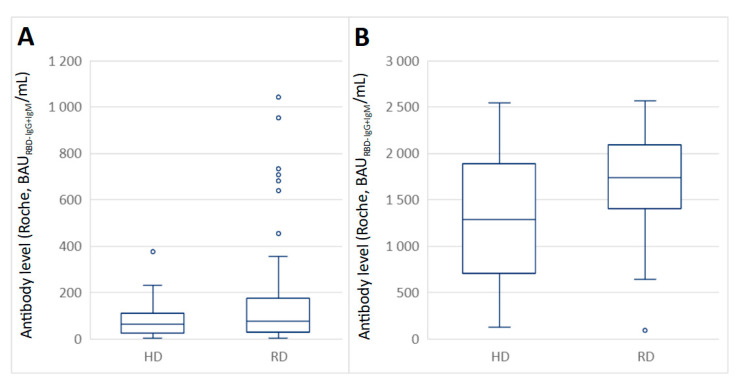
Boxplot (median, hinges: first and third quartiles, whiskers: the largest value no further than 1.5 * IQR from the hinge) showing humoral response to the vaccination as measured with antibody levels in the RD- and HD-COVID-19-negative subgroups on the day of the second dose (FD: *n* = 152, HD: *n* = 33; (**A**) and 8–10 days after the second dose for participants whose results were within the test detection limit (FD: *n* = 45, HD: *n* = 14; (**B**)).

**Table 1 vaccines-09-01277-t001:** Characteristic of participants.

Variables	Number of Participants	Analyzed GroupMale/Female	Mean Age	Mean BMI
RD group (*n* = 188)				
Positive	36	9/27	38.9 (10.9)	24.8 (4.7)
Negative	152	27/125	38.7 (10.5)	24.3 (5.2)
HD group (*n* = 42)				
Positive	9	3/6	35.1 (9.0)	25.0 (4.8)
Negative	33	19/14	39.3 (8.8)	23.8 (4.6)

**Table 2 vaccines-09-01277-t002:** The inclusion and exclusion criteria of HD and RD groups of BNT162b2 vaccinated people under 55 years of age.

Inclusion Criteria	Exclusion Criteria
Age between 18 and 55 years old	Diabetes
Willingness to participate in the study	Hypertension
	Heart-disease
	Chronic pulmonary disease
	Severe allergies
	Obesity (BMI > 30)

## Data Availability

The data presented in this study are available on request from the corresponding author.

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
