# Peer review of "Humoral Response after Vaccination with Half-Dose of BNT162b2 in Subjects under 55 Years of Age"

_vaccines, 2021, doi:10.3390/vaccines9111277_

Round 1

Reviewer 1 Report

Report on the paper

 Using half dose of BNT162b2 vaccine in people under 55 years of age produces comparable amounts of binding antibodies as the full dose - proof of concept.

This interesting manuscript shows that the main problem to solve for the successful fight against COVID-19 is extremely difficult and complex, involving not only the hesitancy to vaccinations but also the vaccination costs. The fate of the entire world seems to range from the optimistic condition of elimination to the conflagration-like state, already affecting India. In between, for countries, like United Kingdom, United States and China, we observe the condition of co-habitation that seems to represent the inability of humankind to win the battle against COVID-19. The reduction of vaccination cost is undoubtedly of great importance to move from co-habitation to elimination.

The result of the experiment done by the authors of this manuscript that half dose of the currently used vaccines is enough to generate antibodies is extremely important, because it corresponds to a strong reduction of the vaccination cost, thereby affording a way to the low-income countries to not fall in the conflagration-like state.

I do not have hesitation to recommend the publication of this manuscript in the present form. The earlier is published, the better.

Reviewer 2 Report

This study aims to confirm that in adults up to 55 years old, the humoral response to the half a dose of BNT162b2 vaccine and to the standard dose (two-dose three-week interval schedule) would be comparable. Based on the results of the present study, the difference in proportions between the participants who did and did not exceed the upper detection limit 8-10 days after dose-2 was not statistically significant. The authors suggest that a half-dose schedule can help limit achieve wide vaccination coverage more quickly and cheaply.

The major concern of this paper is related to the monitoring time. It should be important to evaluate the presence of antibodies in the middle long term. Indeed, although the study demonstrated that the Ab levels could be comparable, the authors should demonstrate that the protection is about the same. In other words, it is important to monitor the subject at different times (after 1/3/6 months) to achieve a general conclusion. These considerations should be inserted in the discussion section as limitations.

In the introduction section, the authors missed inserting several references. Indeed, several paragraphs were reported without a relative bibliography. Please, check it. Moreover, I suggest erasing several unnecessary paragraphs, inserting relevant information that is functional to the study's aims.

The Material and Methods section should be improved. It is important to describe the groups' composition, inserting several missed information such as middle age, sex, BMI, pathologies: for example, it is not clear the groups' composition. Moreover, the inclusion and exclusion criteria should be clearly added. Finally, it could be useful to insert a figure, summarizing the protocol's study. 

The results section should be improved. The results should be reported clearly. Figure 1 is not reported in the main text. Please, check it. Moreover, I suggest inserting as a supplementary file, a new table of the values that have produced the box plot graph. 

The discussion section should be improved. The authors re-present the results in discursive form, while they should compare their results with international data. Please, re-write this section, inserting a section limitation.

Reviewer 3 Report

The title of the research is not concise. Make it more concise and research specific. Remove 'proof of concept' from the title. This is very ambitious and during the peer review stage, it should not be considered as a highly scientific paper.

Page 1 line 46 to 49, expert says that but only one academic source is used. Can you provide any other proof to show that it will take two years? I think this has to be updated.  The poorer countries are also getting the supplies. Hence, it is a request to the authors to update the literature.

The introduction somewhat deviated from the main focus of the research. Is it vaccine hesitancy you wanted to address? 

Mention in the abstract clearly that you recruited participants between 18 to 55 years.

Present the participant recruitment using a flow chart.

Show the inclusion and exclusion criteria in a tabular format.

Show some statistical findings using tables.

Round 2

Reviewer 2 Report

Following the reviewers' comments, the authors have improved the manuscript. I endorse the publication in its current form.

Reviewer 3 Report

Thanks for making all the necessary changes.

This manuscript is a resubmission of an earlier submission. The following is a list of the peer review reports and author responses from that submission.